# Association of *IL18* genetic polymorphisms with Chagas disease in Latin American populations

Mariana Strauss[1]*, Marialbert Acosta-Herrera[2], Alexia Alcaraz[2], Desiré Casares-Marfil[2], Pau Bosch-Nicolau[3], María Silvina Lo Presti[1], Israel Molina[3], Clara Isabel González[4], Chagas Genetics CYTED Network[¶], Javier Martín[2]*

**1** Centro de Estudios e Investigación de la Enfermedad de Chagas y Leishmaniasis, FCM, INICSA-CONICET-UNC, Córdoba, Argentina, **2** Instituto de Parasitología y Biomedicina López-Neyra, IPBLN-CSIC, PTS Granada, Granada, España, **3** Unidad de Medicina Tropical y Salud Internacional Hospital Universitari Vall d'Hebron, PROSICS, Barcelona, España, **4** GIEM, Universidad Industrial de Santander, Bucaramanga, Colombia

¶ Membership of the Chagas Genetics CYTED Network is provided in S1 Membership.
* marianastr86@gmail.com (MS); javiermartin@ipb.csic.es (JM)

**Data Availability Statement:** All relevant data are within the manuscript and its Supporting Information files.

## Abstract

Host genetic factors have been suggested to play an important role in the susceptibility to Chagas disease. Given the influence of interleukin 18 (IL-18) in the development of the disease, in the present study, we analyzed three *IL18* genetic variants (rs2043055, rs1946518, rs360719) regarding the predisposition to *Trypanosoma cruzi* infection and the development of chronic Chagas cardiomyopathy (CCC), in different Latin America populations. Genetic data of 3,608 patients from Colombia, Bolivia, Argentina, and Brazil were meta-analyzed to validate previous findings with increased statistical power. Seropositive and seronegative individuals were compared for *T. cruzi* infection susceptibility. In the Colombian cohort, the allelic frequencies of the three variants showed a significant association, with adjustment for sex and age, and also after applying multiple testing adjustments. Among the Colombian and Argentinean cohorts, rs360719 showed a significant genetic effect in a fixed-effects meta-analysis after a Bonferroni correction (OR: 0.76, CI: 0.66–0.89, P = 0.001). For CCC, the rs2043055 showed an association with protection from cardiomyopathy in the Colombian cohort (OR: 0.79, CI: 0.64–0.99, P = 0.037), with adjustment for sex and age, and after applying multiple testing adjustments. The meta-analysis of the CCC *vs.* asymptomatic patients from the four cohorts showed no evidence of association. In conclusion, our results validated the association found previously in the Colombian cohort suggesting that *IL18* rs360719 plays an important role in the susceptibility to *T. cruzi* infection and no evidence of association was found between the *IL18* genetic variants and CCC in the Latin American population studied.

## Author summary

Chagas disease is a parasitic infection caused by the protozoon *Trypanosoma cruzi*, is the third most common parasitic infection worldwide, the most important in Latin America

**Funding:** This research was supported by grants from Ministerio de Ciencia y Tecnología de Córdoba (GRFT 2017, https://mincyt.cba.gov.ar/), Secretaría de Ciencia y Tecnología, Universidad Nacional de Córdoba, Argentina (https://www.unc.edu.ar/ciencia-y-tecnología/) and Red Iberoamericana de medicina genómica en enfermedad de Chagas - CYTED (http://www.cyted.org). Mariana Strauss performed the experimental work in this article during an internship at the Instituto de Parasitología y Biomedicina López-Neyra, IPBLN-CSIC, Granada, España. The funders had no role in study design, data collection and analysis, decision to publish, or preparation of the manuscript.

**Competing interests:** The authors have declared that no competing interests exist.

and is an emerging disease in non-endemic countries. Actually, millions of people live in areas of exposure and are at risk of contracting the infection. Most of the infected individuals remain asymptomatic for all of their lives, but around 30% of the chronically infected individuals develop irreversible cardiac damage and 10% digestive lesions. Host genetic factors have been suggested to play an important role in the susceptibility or resistance to Chagas disease. In this work, we investigated variants of the *IL18* gene in different Latin America populations. This gene encoded the interleukin-18, which is involved in the immune response to intracellular pathogens like *T. cruzi*. Our results showed that an *IL18* gene variant plays an important role in the protection against *T. cruzi* infection.

## Introduction

Chagas disease is an intracellular and hematic disease caused by the parasite *Trypanosoma cruzi*. Around 6 to 7 million people are estimated to be infected worldwide, most of them being in the poorest rural and urban areas of Latin America, where is endemic [1, 2]. Nowadays, large-scale migrations to other countries have turned Chagas disease into a global health problem [1].

Chagas disease clinical course includes an acute and a chronic phase. The acute phase is characterized by an increase of parasitic load in blood. In this stage, the parasitic load is controlled by the activation of the innate immune response by Th1 pro-inflammatory cytokines such as tumor necrosis factor α (TNF) and interferon γ (IFN-G) [3]. IFN-G activates phagocytic cells to destroy intracellular parasites by inducing TNF and nitric oxide (NO) production [4]. After 8–12 weeks from the infection, individuals evolve into the chronic phase of the disease, in which most of them remain asymptomatic for the rest of their lives. However, around 30–40% of chronically infected patients can develop cardiomyopathy or/and megaviscera. The cardiac involvement is the most frequent manifestation of the disease that occurs in 14–45% of chronically infected patients and affects mainly the conduction system and myocardium [5].

Host genetic factors have been suggested to play an important role in the susceptibility to Chagas disease. [6–9]. Polymorphism in genes encoding cytokines may influence the level of cytokines production and, consequently, cause different immunological responses [10, 11]. Interleukin-18 (IL-18), a pro-inflammatory cytokine produced mainly by macrophages, has been proposed to influence the development of Chagas disease. This cytokine is involved in both innate and adaptive immune response and induces IFN-G production by T cells and NK cells, promoting the Th1 response [12]. Previous genetic studies performed in a Brazilian and Colombian cohort found associations between variants of *IL18* gene and the predisposition to *T. cruzi* infection and chronic Chagas cardiomyopathy [13, 14].

Given the important role played by IL-18 in Chagas disease, in the present study we analyzed the association of three *IL18* genetic variants with the predisposition to *T. cruzi* infection and the development of Chagas cardiomyopathy in different Latin America populations.

## Materials and methods

### Study design and patient population

A candidate-gene case-control study was performed in Colombian, Bolivian and Argentinian cohorts in order to replicate previous findings [13,14]. Additionally, a meta-analysis was performed combining these cohorts.

A total of 3,608 individuals from Latin American countries (Colombia, Bolivia, Argentina and Brazil) were studied. In all cohorts, patients were classified as seropositive for *T. cruzi* antigens (n = 2,890) and seronegative (n = 718) based on results of at least 2 of 3 independent tests. Based on clinical evaluation, an electrocardiogram and echocardiogram were recorded to detect any conduction and structural alterations. Subsequently, the seropositive patients who presented cardiac alterations were classified as chronic Chagas cardiomyopathy (CCC, n = 1,707) and asymptomatic (ASY, n = 1,183). The sex distribution for the entire Latin American population studied was 61.4% female and 38.6% male.

**Colombian cohort.** A total of 406 Colombian individuals from the same population as the study by Leon Rodriguez et al. (2016) [14] were recruited by the health care team from the Industrial University of Santander and Cardiovascular Foundation from Colombia. In order to increase the sample size, these individuals were included with the previously published Colombian cohort, making a total of 1,577 individuals. From this, 937 were classified as seropositive for *T. cruzi* antigens and 640 were classified as seronegative (according to the serological tests: recombinant antigen ELISA and commercial indirect hemagglutination test). Based on complementary tests and clinical findings, seropositive patients were classified as CCC = 576 and ASY = 361. The mean age of participants was 45.55 ± 17.19 years for seronegative individuals, CCC = 61.44 ± 12.82 and ASY = 51.90 ± 14.18. The sex distribution was 58% female and 42% male.

**Bolivian cohort.** A total of 630 Bolivian individuals residents in Barcelona, Spain were recruited from the Infectious Diseases Department of the Vall d'Hebron University Hospital. In this cohort, only seropositive patients were classified as CCC = 100 and ASY = 530 based on complementary tests and clinical findings. The mean age of the participants was CCC = 50.71 ± 9.41 and ASY = 46.93 ± 9.49. The sex distribution was 69% female and 31% male.

**Argentinian cohort.** A total of 350 Argentinian individuals from an endemic region for Chagas disease (Cordoba province) were included in this study. The study subjects were recruited from the National Hospital of Clinics and Sucre Clinic, Cordoba city. The population in this region of Argentina is a homogeneous mixture, with no specific concentration of any ethnicity. All participants underwent a serological diagnosis for *T. cruzi* infection through the enzyme-linked immunosorbent assay (ELISA) that uses recombinant antigen and a commercial indirect hemagglutination test. According to the results of these tests, 272 individuals were classified as seropositive for *T. cruzi* antigens and 78 were classified as seronegative. Based on the results of complementary tests and clinical findings, seropositive patients were classified as CCC = 182 and ASY = 90. The mean age of participants was 53.82 ± 16.53 years for seronegative individuals, CCC = 60.14 ± 10.16 and ASY = 49.30 ± 13.65. The sex distribution was 71% female and 29% male.

**Brazilian cohort [13].** A total of 1,051 Brazilian seropositive patients for antibodies against *T. cruzi* were included in the meta-analysis. From this, 849 individuals were classified as CCC and 202 ASY. The sex distribution was 52% female and 48% male.

## Ethics statement

The study was accepted by the Ethics Committees from the Industrial University of Santander and Cardiovascular Foundation, Colombia; the Vall d'Hebron University Hospital, Barcelona, Spain and the National Hospital of Clinics, National University of Cordoba, Argentina. Written informed consent was obtained from all subjects prior to participation. The research protocols followed the principles of the Declaration of Helsinki and informed consent was obtained from all individual participants included in the study.

## Selected polymorphisms and genotyping

The gene encoding IL-18 is located on chromosome 11q22.2-q22.3 [15] and consists of six exons and five introns [16] (**S1A Fig**). Three SNPs previously studied in Chagas disease were selected: rs2043055, rs1946518 and rs360719 for this study [13, 14]. Linkage disequilibrium (LD, $R^2$ and D') was estimated using an expectation–maximization algorithm implemented in Haploview V4.2 [17] for the studied cohorts: Colombian, Bolivian, Argentinian, and from the American sub-populations genotype data from the 1000 Genomes Project phase III (http://www.1000genomes.org) [18].

Genomic DNA from blood samples was isolated following standard procedures and the genotyping was performed using TaqMan assays (Applied Biosystems, Foster City, California, USA) on a real-time PCR system (7900HT Fast Real-Time PCR System), SNPs were determined by TaqMan 5′ allelic discrimination assay method performed by Applied Biosystems.

## Statistical analysis

For the candidate gene study, the statistical analyses were performed with the software Plink V1.9 (http://zzz.bwh.harvard.edu/plink/plink2.shtml) [19]. Deviance from Hardy-Weinberg equilibrium was determined at the 1% significance level in all groups of individuals. Individuals that have not achieved an SNP completion rate of 95% have been filtered out. To test for possible allelic, logistic regression model and Fisher's exact test were assessed in seronegative *vs*. seropositive individuals and asymptomatic *vs*. chronic Chagas cardiomyopathy individuals. The Benjamini & Hochberg step-up false discovery rate (FDR) correction was used in all analyses to control for multiple testing. The covariates sex and age were adjusted in logistic regression models. P-values lower than 0.05 were considered as statistically significant.

To assess the consistency of effects across the cohorts, a meta-analysis was performed with METASOFT (http://genetics.cs.ucla.edu/meta/) based on inverse-variance-weighted effect size. Heterogeneity across studies was assessed using Cochran's Q statistic (Q test $P < 0.05$) and $I^2$ heterogeneity index [20]. A fixed-effects model was applied for those SNPs without evidence of heterogeneity (Cochran's Q test $P > 0.05$), and a random-effects model was applied for SNPs displaying heterogeneity of effects between studies (Cochran's Q test $P \leq 0.05$). The significance threshold for the meta-analyses was estimated based on the Bonferroni correction ($0.05/3 = 0.017$) [21].

The statistical power of the studies was estimated with the Power Calculator for Genetic Studies 2006 (CaTS) software (**S1 Table**) (http://www.sph.umich.edu/csg/abecasis/CaTS/) [22].

Evaluation of functionality of the three SNPs analyzed was performed with the online software HaploReg v4.1 [23] (https://pubs.broadinstitute.org/mammals/haploreg/haploreg.php) based on empirical data from the ENCODE project (http://www.genome.gov/encode/). Specifically, we focused our attention on experiments performed on blood and T cells lines in the American population. For regulatory features, Ensembl Browser (https://www.ensembl.org) [18] and ReMap 2018 v1.2. (http://tagc.univ-mrs.fr/remap/) [24] were used.

## Results

The three *IL18* SNPs were in Hardy-Weinberg equilibrium in all the analyzed cohorts ($P > 0.01$). The genotyping success rate was over 90% and the allele frequencies in all cases were similar to those described for the Americans sub-populations of the 1000 Genomes Project phase III [18] (**S1 Table**). The SNPs were in moderate pairwise linkage disequilibrium in the studied cohorts and in the American sub-populations from the 1000 Genomes Project phase III [18] (**S1B and S1C Fig**).

**Table 1. Colombian cohort. Genotype and allele distribution for *IL18* variants in seronegative and seropositive individuals.**

| SNP | | A1\| A2 | Genotype. N (%) | | | MAF | Allele test | | | |
|---|---|---|---|---|---|---|---|---|---|---|
| | | | 1\|1 | 1\|2 | 2\|2 | | OR | (L95-U95) | P LogstReg | P FDR_BH |
| rs2043055 | Seronegative (631) | G\|A | 82 (13.00) | 300 (47.54) | 249 (39.46) | 36.77 | 1.30 | (1.10–1.53) | **0.002** | **0.004** |
| | Seropositive (927) | | 164 (17.69) | 450 (48.54) | 313 (33.76) | 41.96 | | | | |
| rs1946518 | Seronegative (631) | T\|G | 163 (25.83) | 334 (52.93) | 134 (21.24) | 52.3 | 0.79 | (0.67–0.92) | **0.003** | **0.004** |
| | Seropositive (927) | | 214 (23.09) | 448 (48.33) | 265 (28.59) | 47.25 | | | | |
| rs360719 | Seronegative (631) | G\|A | 103 (16.32) | 299 (47.39) | 229 (36.29) | 40.02 | 0.75 | (0.63–0.89) | **0.001** | **0.004** |
| | Seropositive (927) | | 107 (11.54) | 426 (45.95) | 394 (42.50) | 34.52 | | | | |

1: minor allele | 2: major allele; alleles are showed in forward strand. MAF: minor allele frequency. OR: odds ratios, L95-U95: confidence intervals of 95% L: lower limit; U: upper limit. Values adjusted by sex and age. Significant P value is shown in bold.

## *T. cruzi* infection susceptibility

The allelic and genotypic frequencies of seronegative and seropositive individuals from Colombia were compared in **Table 1**. The allelic frequencies of the three SNPs were statistically significant even after multiple testing correction. The frequency of the minor allele, G, in rs2043055 was significantly reduced in the seronegative compared to seropositive individuals suggesting an association with higher infection risk, while the frequencies of rs1946518*T and rs360719*G alleles were significantly increased in seronegative compared to the seropositive individuals, suggesting an association with the protection against the infection by *T. cruzi*.

The allelic and genotypic frequencies of seronegative and seropositive individuals from Argentina are shown in **Table 2**. No associations between *IL18* genetic variants were found. However, the rs2043055 remained borderline significant for protection against infection by *T. cruzi* [P = 0.061, odds ratio (OR) = 0.71, 95% confidence interval (CI) = 0.49–1.02].

In addition, a meta-analysis combining data from Colombian and Argentinean cohorts was performed (**Table 3**). The *IL18* rs360719 showed consistent effects among the two meta-analyzed populations with a statistically significant association (CI: 0.66–0.89, P = 0.001, under a fixed-effects meta-analysis) with an OR for the G allele of 0.76, which was statistically significant after a Bonferroni correction (P< 0.017). For this comparison, the sample size attained a statistical power of over 80% for this OR (**S1B1 Table**). In both cohorts, the allele effects size were in concordance and this result indicates an association to the protection against *T. cruzi* infection in these cohorts.

**Table 2. Argentinian cohort. Genotype and allele distribution for *IL18* variants in seronegative and seropositive individuals.**

| SNP | | A1\| A2 | Genotype. N (%) | | | MAF | Allele test | | |
|---|---|---|---|---|---|---|---|---|---|
| | | | 1\|1 | 1\|2 | 2\|2 | | OR | (L95-U95) | P LogstReg |
| rs2043055 | Seronegative (77) | G\|A | 20 (25.97) | 29 (37.66) | 28 (36.36) | 44.81 | 0.71 | (0.49–1.02) | 0.061 |
| | Seropositive (270) | | 33 (12.23) | 126 (46.67) | 111 (41.11) | 35.56 | | | |
| rs1946518 | Seronegative (77) | T\|G | 19 (24.67) | 35 (45.46) | 23 (29.87) | 47.4 | 1.03 | (0.71–1.49) | 0.883 |
| | Seropositive (270) | | 54 (20) | 151 (55.92) | 65 (24.08) | 47.96 | | | |
| rs360719 | Seronegative (77) | G\|A | 11 (14.28) | 33 (42.85) | 33 (42.85) | 35.71 | 0.87 | (0.60–1.31) | 0.552 |
| | Seropositive (270) | | 25 (9.26) | 128 (47.40) | 117 (43.33) | 32.96 | | | |

1: minor allele | 2: major allele; alleles are showed in forward strand. MAF: minor allele frequency. OR: odds ratios, L95-U95: confidence intervals of 95% L: lower limit; U: upper limit. Values adjusted by sex and age.

**Table 3. Meta-analysis of *IL18* variants, Argentinian and Colombian cohorts for *T. cruzi* infection susceptibility.**

| SNP | Colombian cohort | | | Argentinian cohort | | | Meta-analysis | | |
|---|---|---|---|---|---|---|---|---|---|
| | OR | (L95-U95) | P | OR | (L95-U95) | P | OR | (L95-U95) | P |
| rs2043055 | 1.30 | (1.10–1.53) | 0.002 | 0.71 | (0.49–1.02) | 0.061 | 1.17 | (1.01–1.36) | 0.035 |
| rs1946518 | 0.79 | (0.67–0.92) | 0.003 | 1.03 | (0.71–1.49) | 0.883 | 0.82 | (0.71–0.94) | 0.006 |
| rs360719 | 0.75 | (0.63–0.89) | 0.001 | 0.87 | (0.60–1.31) | 0.552 | 0.76 | (0.66–0.89) | **0.001** |

Total number of individuals: seropositive n = 1,209 and seronegative n = 718

OR: odds ratios, L95-U95: confidence intervals of 95% L: lower limit; U: upper limit. Marked in bold the P value ≤ than the individual cohorts. Significant association based on the Bonferroni correction P< 0.017.

## Chronic Chagas cardiomyopathy susceptibility

The allelic and genotypic frequencies of chronic Chagas cardiomyopathy and asymptomatic patients from Colombia were compared in **Table 4**. The allelic frequencies of *IL18* rs2043055 was statistically significant even after multiple testing correction (P = 0.037, OR = 0.79, CI = 0.64–0.99). The frequency of the of the rs2043055* G allele was significantly incremented in asymptomatic patients, suggesting an association with the protection against the development of Chagas cardiomyopathy. However, no significant differences in allelic frequencies were observed for rs1946518 and rs360719.

The SNP *IL18* rs2043055 was studied in 1,051 seropositive Brazilian patients (CCC = 849 and ASY = 202) [13]. The frequency of the G allele, in rs2043055, was increased in chronic Chagas cardiomyopathy patients compared to asymptomatic in the Brazilian, Bolivian and Argentinian cohorts, but no significant differences were found in these cohorts. Nevertheless, in the Bolivian cohort (**Table 5**), a trend of association can be observed for this SNP (P = 0.088, OR = 1.39, CI = 0.95–2.02). In the Argentinian cohort (**Table 6**), the frequency of the T allele, in *IL18* rs1946518, was increased in asymptomatic compared to chronic Chagas cardiomyopathy patients, and remained borderline significant for suggesting an association with the protection against the development of Chagas cardiomyopathy (P = 0.078, OR = 0.67, CI = 0.44–1.04).

Further, a meta-analysis combining these results were performed (**Table 7**). The results of the available SNPs showed no significant associations.

## In silico functional characterization of *IL18* gene variants

We further explored the functional annotations of the three variants studied in this work using HaploReg v4.1. The annotation indicated that the SNPs of *IL18* are located in a regulatory

**Table 4. Colombian cohort.** Genotype and allele distribution for *IL18* variants in asymptomatic and chronic Chagas cardiomyopathy (CCC) individuals.

| SNP | | A1\| A2 | Genotype. N (%) | | | MAF | Allele test | | | |
|---|---|---|---|---|---|---|---|---|---|---|
| | | | 1\|1 | 1\|2 | 2\|2 | | OR | (L95-U95) | P LogstReg | P FDR_BH |
| rs2043055 | Asymptomatic (358) | G\|A | 83 (23.18) | 159 (44.41) | 116 (32.40) | 45.39 | 0.79 | (0.64–0.99) | **0.037** | **0.046** |
| | CCC (569) | | 81 (14.24) | 291 (51.14) | 197 (34.62) | 39.81 | | | | |
| rs1946518 | Asymptomatic (358) | T\|G | 82 (22.91) | 160 (44.69) | 116 (32.40) | 45.25 | 1.14 | (0.92–1.41) | 0.225 | 0.229 |
| | CCC (569) | | 132 (23.20) | 288 (50.62) | 149 (26.19) | 48.51 | | | | |
| rs360719 | Asymptomatic (358) | G\|A | 45 (12.57) | 155 (43.30) | 158 (44.13) | 34.22 | 0.99 | (0.79–1.26) | 0.994 | 0.765 |
| | CCC (569) | | 62 (10.90) | 271 (47.63) | 236 (41.48) | 34.71 | | | | |

1: minor allele | 2: major allele; alleles are showed in forward strand. MAF: minor allele frequency. OR: odds ratios, L95-U95: confidence intervals of 95% L: lower limit; U: upper limit. Values adjusted by sex and age. Significant P value is shown in bold.

**Table 5. Bolivian cohort.** Genotype and allele distribution for *IL18* variants in asymptomatic and chronic Chagas cardiomyopathy (CCC) individuals.

| SNP | | A1\| A2 | Genotype. N (%) | | | MAF | Allele test | | |
|---|---|---|---|---|---|---|---|---|---|
| | | | 1\|1 | 1\|2 | 2\|2 | | OR | (L95-U95) | P LogstReg |
| rs2043055 | Asymptomatic (528) | G\|A | 72 (13.64) | 260 (49.24) | 196 (37.12) | 38.26 | 1.39 | (0.95–2.02) | 0.088 |
| | CCC (100) | | 16 (16) | 50 (50) | 34 (34) | 41 | | | |
| rs1946518 | Asymptomatic (528) | G\|T | 101 (19.13) | 268 (50.76) | 159 (30.11) | 44.51 | 1.24 | (0.85–1.80) | 0.260 |
| | CCC (100) | | 19 (19) | 55 (55) | 26 (26) | 46.5 | | | |
| rs360719 | Asymptomatic (528) | G\|A | 54 (10.23) | 237 (44.89) | 237 (44.89) | 32.67 | 0.98 | (0.66–1.45) | 0.934 |
| | CCC (100) | | 10 (10) | 50 (50) | 40 (40) | 35 | | | |

1: minor allele | 2: major allele; alleles are showed in forward strand. MAF: minor allele frequency. OR: odds ratios, L95-U95: confidence intervals of 95% L: lower limit; U: upper limit. Values adjusted by sex and age.

region of the genome (S2 Table). The annotation based on the epigenomic information of rs2043055 indicates that this SNP maps in an enhancer region, which is correlated with active gene expression in primary mononuclear cells and in T cells from peripheral blood. The rs1946518 and rs360719 variants mapped in a region enriched in histone marks: H3K4me3, H3K9ac, a hallmark of active promoter region, and H3K27ac in enhancer region in mononuclear cells and T cells (S2A Table). Furthermore, according to ReMap 2018 v1.2 these three SNPs mapped in regulatory regions of the human genome and it has been described as transcription factors (S2B Table).

## Discussion

Genetic factors and immunologic response may determine the susceptibility against the infection and development of Chagas disease [6–9]. In the present study, three *IL18* genetic variants were analyzed in four Latin American populations, being the largest genetic study conducted to date in Chagas disease. Concerning genetic control of the infection, our results evidenced the implication of the *IL18* rs360719 polymorphism, in our populations. However, when comparing cardiomyopathy and asymptomatic patients, no significant associations were detected.

Addressing the question of genetic susceptibility to *T. cruzi* infection through comparison of seropositive with seronegative individuals is not an easy task. The recruitment of an adequate number of subjects from endemic areas exposed to *T. cruzi* is often challenging, and that is the reason for including only two cohorts in this comparison. The previous study in a Colombian cohort was the first to report an association for rs2043055, rs1946518 and rs360719 with *T. cruzi* infection and suggested that this association was mainly driven by the

**Table 6. Argentinian cohort.** Genotype and allele distribution for *IL18* variants in asymptomatic and chronic Chagas cardiomyopathy (CCC) individuals.

| SNP | | A1\| A2 | Genotype. N (%) | | | MAF | Allele test | | |
|---|---|---|---|---|---|---|---|---|---|
| | | | 1\|1 | 1\|2 | 2\|2 | | OR | (L95-U95) | P LogstReg |
| rs2043055 | Asymptomatic (89) | G\|A | 9 (10.11) | 40 (44.94) | 40 (44.94) | 32.58 | 1.26 | (0.82–1.95) | 0.291 |
| | CCC (181) | | 24 (13.26) | 86 (47.51) | 71 (39.22) | 37.02 | | | |
| rs1946518 | Asymptomatic (89) | T\|G | 21 (23.59) | 51 (57.30) | 17 (19.10) | 52.25 | 0.67 | (0.44–1.04) | 0.078 |
| | CCC (181) | | 33 (18.23) | 100 (55.25) | 48 (26.51) | 45.86 | | | |
| rs360719 | Asymptomatic (89) | G\|A | 8 (8.99) | 47 (52.81) | 34 (38.20) | 35.39 | 0.81 | (0.52–1.27) | 0.364 |
| | CCC (181) | | 17 (9.32) | 81 (44.75) | 83 (45.86) | 31.77 | | | |

1: minor allele | 2: major allele; alleles are showed in forward strand. MAF: minor allele frequency. OR: odds ratios, L95-U95: confidence intervals of 95% L: lower limit; U: upper limit. Values adjusted by sex and age.

**Table 7. Meta-analysis of *IL18* variants, Latin American cohorts for Chagas cardiomyopathy susceptibility.**

|  | Colombian cohort | | Bolivian cohort | | Argentinian cohort | | Brazilian cohort | | Meta-analysis | |
|---|---|---|---|---|---|---|---|---|---|---|
| SNP | OR (L95-U95) | P | OR (L95-U95) | P | OR (L95-U95) | P | OR (L95-U95) | P | OR (L95-U95) | P |
| rs2043055 | 0.79 (0.64–0.99) | 0.037 | 1.39 (0.95–2.02) | 0.088 | 1.26 (0.82–1.95) | 0.291 | 1.06 (0.85–1.32) | 0.598 | 1.05 (0.82–1.35) | 0.259 |
| rs1946518 | 1.14 (0.92–1.41) | 0.225 | 1.24 (0.85–1.80) | 0.260 | 0.67 (0.44–1.04) | 0.078 | - | - | 1.07 (0.90–1.26) | 0.426 |
| rs360719 | 0.99 (0.79–1.26) | 0.994 | 0.98 (0.66–1.45) | 0.934 | 0.81 (0.52–1.27) | 0.364 | - | - | 0.95 (0.79–1.15) | 0.629 |

Total number of individuals: rs2043055 CCC n = 1,707 and asymptomatic n = 1,183; rs1946518 and rs360719: CCC n = 858 and asymptomatic n = 981

OR: odds ratios, L95-U95: confidence intervals of 95% L: lower limit; U: upper limit.

polymorphism rs360719 [14]. After the enlargement of this cohort, the association remained, showing consistent results in a well-powered cohort. Replication of these variants was performed in an Argentinian cohort and only rs2043055, showed a borderline genetic association but in the opposite direction compared with the Colombian cohort. These differences could be due to the complex genetic structure of Latin American individuals, reflected by the recent admixture among Native American, European, and West African source populations [25]. Also, the lack of replication may occur if the assessed polymorphism is not the causal variant but is rather in LD with it, i.e., variants correlated with each other more often than expected by chance. LD patterns depend on the genetic background of the founder population and population history [26]. The rs360719 in the Argentinian cohort showed no association with *T. cruzi* infection, which could be a consequence of an insufficient statistical power (S1**A1 Table**). Interestingly, the meta-analysis showed that the variant *IL18* rs360719 presented a consistent effect among the two cohorts, indicating protection against *T. cruzi* infection. The SNP rs360719 is located in the promoter region of the *IL18* gene. The functional annotation of this SNP with empirical data from the ENCODE project revealed that is located in histone marks in primary mononuclear cells and T helper naive cells from peripheral blood, and has been described as transcription factor (S2 **Table**). These modifications are critically involved in the regulation of gene expression [27]. Also, it has been described that *IL18* rs360719 polymorphism leads to loss of the octamer (OCT)-1 transcription factor binding site. OCT-1 is known to be a ubiquitously expressed factor involved in the regulation of certain cytokines, like IL-18 [28]. Thus, the rs360719 would be associated with IL-18 expression in peripheral blood mononuclear cells and may play a role in the susceptibility or resistance to *T. cruzi* infection.

Chronic chagasic cardiomyopathy, the most frequent clinical outcome of Chagas disease, has been associated with cytokine enriched heart tissue inflammation [29]. Furthermore, local expression of IL-18 in chronic chagasic cardiomyopathy heart tissue has been described and would be associated with mononuclear inflammatory infiltrates, cardiomyocyte destruction and fibrosis [30]. In our study, we analyzed *IL18* gene variants in four Latin American populations with chronic Chagas cardiomyopathy. The *IL18* genetic variant, rs2043055, studied in the Brazilian cohort, showed nominal significant differences in the genotypic frequencies among moderate and severe chagasic cardiomyopathy patients [13]. When comparing chronic chagasic cardiomyopathy with asymptomatic patients, this variant showed a significant association in the Colombian cohort. However, these results were not validated in the Bolivian and Argentinian cohorts. These discrepancies in the results could be due to the genetic heterogeneity among the study cohorts [25]. The impact of European colonization and slave trade from western Africa has altered the genomes of Native Americans in multiple and dynamic ways. Approximately, 9–9.6 generations have passed since admixture and ancestry-enriched SNPs in Latin American populations may have a substantial effect on health and disease related phenotypes [31, 32]. For instance, Norris et al. reported SNPs with excess African or European

ancestry, which are associated with ancestry-specific gene expression patterns and play crucial roles in the immune system and infectious disease responses [33]. In addition, an interesting report by Lima-Costa et al. showed that the prevalence of *T. cruzi* infection is strongly and independently associated with higher levels of African and Native American ancestry in a Brazilian population [34]. This heterogeneity in ancestry proportions across geographic regions, and also within countries themselves, are challenging in association studies in order to find generalizable results across populations [35–37]. All this, suggests that a fine-scale genomics perspective might represent a powerful tool to understand the role of genetics in this neglected disease diagnosis and prognosis.

As was mentioned, IL-18 plays an important role in the regulation of IFN-G production and development of Th1 response. This interleukin is produced by a wide variety of cells, including dendritic cells, macrophages, keratinocytes, intestinal epithelial cells, and osteoblasts, suggesting a key pathophysiological role in health and disease [12]. Several studies have highlighted the implication of IL-18 in the acute and chronic phase of Chagas disease [38–41]. Considering that infectious diseases exert significant selective genetic pressure, it has been postulated two genetic mechanisms to explain the pathogenesis of Chagas disease [8]. First, *pathogen resistance genes* (PRG) would be involved in inhibits infection by directly reducing pathogen burden and secondly, *disease tolerance genes* (DTG) will operate to minimize tissue damaging effects of the pathogen [42–44]. Consequently, polymorphisms in PRG and DTG will be associated with differential disease progression. One of the most relevant disease tolerance genes identified was related to directly or indirectly inhibit IFN-G production or Th1 differentiation [8], and therefore, IL-18 could be implicated in this regulation.

In conclusion, our results validated the previous work suggesting that *IL18* rs360719 plays an important role in the susceptibility to *T. cruzi* infection [14]. On the other hand, no evidence of association was found between the *IL18* genetic variants and chronic Chagas cardiomyopathy in the Latin American population. Even though, meta-analyses offers a powerful approach to identify genetic variants that influence susceptibility of common diseases [45, 46], in the context of Chagas disease is necessary to contemplate the challenges of studying such an heterogeneous populations like Latin Americans with recent admixture, where fine-scale genomic assessments may be necessary [25]. Finally, further studies are needed to reach more conclusive results concerning the genetic basis of Chagas disease.

## Supporting information

**S1 Fig.** Location of *IL18* rs2043055, rs1946518 and rs360719 within the gen (A). R2 (B) and Linkage disequilibrium D' (C) plots estimated by using expectation maximization algorithmin Haploview V4.2. in Americans (AMR) from 1000 Genomes Project Phase III and in Colombian, Bolivian and Argentinian cohorts.
(TIF)

**S1 Table.** Statistical power calculation of the candidate gene study (A) and the meta-analysis (B) considering the allele frequencies for each SNP, the prevalence of the disease in each country with three different OR.
(DOCX)

**S2 Table.** In silico functional characterization of *IL18* gene variants (A) Regulatory feature consequences for *IL18* gene variants (B).
(DOCX)

**S1 Membership. Membership of the Chagas Genetics CYTED Network.**
(DOCX)

## Acknowledgments

We thank all the patients who participated in this study and the Medical team from the different Latin American countries and Spain.

## Author Contributions

**Conceptualization:** Mariana Strauss, Marialbert Acosta-Herrera, Israel Molina, Javier Martín.

**Data curation:** Mariana Strauss, Alexia Alcaraz, Desiré Casares-Marfil.

**Formal analysis:** Mariana Strauss, Marialbert Acosta-Herrera, Alexia Alcaraz, Desiré Casares-Marfil.

**Funding acquisition:** María Silvina Lo Presti, Israel Molina, Clara Isabel González, Javier Martín.

**Investigation:** Mariana Strauss, Marialbert Acosta-Herrera, Alexia Alcaraz, Pau Bosch-Nicolau, María Silvina Lo Presti, Israel Molina, Clara Isabel González, Javier Martín.

**Methodology:** Mariana Strauss, Marialbert Acosta-Herrera.

**Project administration:** Marialbert Acosta-Herrera, Javier Martín.

**Resources:** Alexia Alcaraz, Pau Bosch-Nicolau, María Silvina Lo Presti.

**Software:** Mariana Strauss, Desiré Casares-Marfil.

**Supervision:** Marialbert Acosta-Herrera, Javier Martín.

**Validation:** Mariana Strauss, Israel Molina, Clara Isabel González, Javier Martín.

**Visualization:** Mariana Strauss, Pau Bosch-Nicolau, María Silvina Lo Presti, Israel Molina, Clara Isabel González, Javier Martín.

**Writing – original draft:** Mariana Strauss, Marialbert Acosta-Herrera, Alexia Alcaraz.

**Writing – review & editing:** Mariana Strauss, Marialbert Acosta-Herrera, Pau Bosch-Nicolau, María Silvina Lo Presti, Israel Molina, Clara Isabel González, Javier Martín.

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
