## [Decision Letter · Decision Letter 0]

14 Aug 2019

Dear Dr Strauss:

Thank you very much for submitting your manuscript "Association of IL18 genetic polymorphisms with Chagas disease in Latin American populations" (#PNTD-D-19-01126) for review by PLOS Neglected Tropical Diseases. Your manuscript was fully evaluated at the editorial level and by independent peer reviewers. The reviewers appreciated the attention to an important problem, but raised some substantial concerns about the manuscript as it currently stands. These issues must be addressed before we would be willing to consider a revised version of your study. We cannot, of course, promise publication at that time.

We therefore ask you to modify the manuscript according to the review recommendations before we can consider your manuscript for acceptance. Your revisions should address the specific points made by each reviewer. 

When you are ready to resubmit, please be prepared to upload the following:

(1) A letter containing a detailed list of your responses to the review comments and a description of the changes you have made in the manuscript.

(2) Two versions of the manuscript: one with either highlights or tracked changes denoting where the text has been changed (uploaded as a "Revised Article with Changes Highlighted" file); the other a clean version (uploaded as the article file).

(3) If available, a striking still image (a new image if one is available or an existing one from within your manuscript). If your manuscript is accepted for publication, this image may be featured on our website. Images should ideally be high resolution, eye-catching, single panel images; where one is available, please use 'add file' at the time of resubmission and select 'striking image' as the file type. 

Please provide a short caption, including credits, uploaded as a separate "Other" file. If your image is from someone other than yourself, please ensure that the artist has read and agreed to the terms and conditions of the Creative Commons Attribution License at http://journals.plos.org/plosntds/s/content-license (NOTE: we cannot publish copyrighted images). 

(4) If applicable, we encourage you to add a list of accession numbers/ID numbers for genes and proteins mentioned in the text (these should be listed as a paragraph at the end of the manuscript). You can supply accession numbers for any database, so long as the database is publicly accessible and stable. Examples include LocusLink and SwissProt.

(5) To enhance the reproducibility of your results, we recommend that you deposit your laboratory protocols in protocols.io, where a protocol can be assigned its own identifier (DOI) such that it can be cited independently in the future. For instructions see http://journals.plos.org/plosntds/s/submission-guidelines#loc-methods

While revising your submission, please upload your figure files to the Preflight Analysis and Conversion Engine (PACE) digital diagnostic tool, https://pacev2.apexcovantage.com/ PACE helps ensure that figures meet PLOS requirements. To use PACE, you must first register as a user. Then, login and navigate to the UPLOAD tab, where you will find detailed instructions on how to use the tool. If you encounter any issues or have any questions when using PACE, please email us at figures@plos.org.

We hope to receive your revised manuscript by Oct 13 2019 11:59PM. If you anticipate any delay in its return, we ask that you let us know the expected resubmission date by replying to this email.

To submit a revision, go to https://www.editorialmanager.com/pntd/ and log in as an Author. You will see a menu item call Submission Needing Revision. You will find your submission record there. 

Sincerely,

Ana Rodriguez

Deputy Editor

Reviewer's Responses to Questions

**Key Review Criteria Required for Acceptance?**

**Methods**

-Are the objectives of the study clearly articulated with a clear testable hypothesis stated?

-Is the study design appropriate to address the stated objectives?

-Is the population clearly described and appropriate for the hypothesis being tested?

-Is the sample size sufficient to ensure adequate power to address the hypothesis being tested?

-Were correct statistical analysis used to support conclusions?

-Are there concerns about ethical or regulatory requirements being met?

Reviewer #1: Strauss and colleagues performed a replication study to confirm whether IL18 variation was associated with predisposition to Trypanosoma cruzi infection and the development of chronic Chagas cardiomyopathy in Latin American populations. With that aim, they recruited a considerable large case-control study group from different countries, which allowed them to conduct the analyses with an adequate statistical power. An appropriate ethical statement was included, and I do not have concerns about regulatory requirements being met, as the study was performed by an international consortium with a proven experience on this matter. However, I have some considerations to make in relation to the methods.

- Although it is understood that the SNP selection criteria was based on a previously published study, it would be interesting to include additional supplementary information about the gene structure and specific location of the analysed genetic variants within the region.

- The demographic and clinical data of the analysed Brazilian cohort was missing. The authors should indicate the number of individuals showing each phenotype and whether the sex and age information was available, as mentioned for the other cohorts.

Reviewer #2: -Are the objectives of the study clearly articulated with a clear testable hypothesis stated? 

YES

-Is the study design appropriate to address the stated objectives?

YES

-Is the population clearly described and appropriate for the hypothesis being tested?

YES

-Is the sample size sufficient to ensure adequate power to address the hypothesis being tested?

YES

-Were correct statistical analysis used to support conclusions?

YES but the authors can go further with additional ones.

-Are there concerns about ethical or regulatory requirements being met?

YES

**Results**

-Does the analysis presented match the analysis plan?

-Are the results clearly and completely presented?

-Are the figures (Tables, Images) of sufficient quality for clarity?

Reviewer #1: The results are consistent and I must say that they have been brilliantly organised and discussed. The workflow was appropriate and the tables clearly presented and with a good quality. I have just one comment related to this section:

- An evaluation of functionality of the three studied SNPs was performed. However, no description of the findings was included in the results section.

Reviewer #2: -Does the analysis presented match the analysis plan?

YES

-Are the results clearly and completely presented?

YES

-Are the figures (Tables, Images) of sufficient quality for clarity?

YES

**Conclusions**

-Are the conclusions supported by the data presented?

-Are the limitations of analysis clearly described?

-Do the authors discuss how these data can be helpful to advance our understanding of the topic under study?

-Is public health relevance addressed?

Reviewer #1: In general, this is a well structured and nicely written manuscript in which the conclusions are clear and supported by the data obtained. This type of studies is of high relevance considering the incidence of Chagas disease in the studied areas and the lack of knowledge regarding the genetic influence in the development of its most severe phenotypes. In this regard, the relevance for the public health is addressed in the manuscript. However, although the authors acknowledge the limitations of the study, I have a specific concern regarding the following:

- The authors speculated that the discrepancies observed in their results could be due to genetic heterogeneity amongst the study populations. I may agree on this, but a more detailed discussion of this idea (using more recent citations) should be added to the manuscript.

Reviewer #2: -Are the conclusions supported by the data presented?

YES

-Are the limitations of analysis clearly described?

+/-

-Do the authors discuss how these data can be helpful to advance our understanding of the topic under study?

+/-

-Is public health relevance addressed?

YES

**Editorial and Data Presentation Modifications?**

Reviewer #1: The supplementary tables are differently called in the text and in the supporting information (e.g. Table S1 / Table A).

Reviewer #2: Association of IL18 genetic polymorphisms with Chagas disease in Latin American populations.

Reference PNTD-D-19-01126

Given the influence of interleukin 18 in the development of chagas disease, the authors analyzed three IL18 variants regarding the predisposition to Trypanosoma cruzi infection and the development of chronic Chagas cardiomyopathy (CCC), in several Latin America populations. 

1) Genetic data were obtained for 3,608 patients from Colombia, Bolivia, Argentina, and Brazil. Authors performed meta-analysis to validate previous findings with increased statistical power.

2) Among the Colombian and Argentinean cohorts, rs36071927 showed a significant genetic effect.

3) rs2043055 showed an association with protection from cardiomyopathy in the Colombian cohort.

4) The meta-analysis of the CCC vs asymptomatic patients from the four cohorts showed no evidence of association.

According to the authors their results validated the association found previously in the Colombian cohort suggesting that IL18 rs360719 plays an important role in the susceptibility to T. cruzi infection and no evidence of association was found between the IL18 genetic variants and CCC in the Latin American population studied.

Major corrections:

Page 6 126-127: Three single nucleotide polymorphisms (SNPs) of IL18 gene, rs2043055, rs1946518 and rs360719 were selected. The selected SNPs were in moderate pairwise linkage disequilibrium (LD, r2 < 0.5) in the American population; values were estimated using LDlink website tool 

The authors need to provide in supplementary data the D’ and R2 values between each marker in their various study populations and in the reference population. Moreover, there is not enough information on the American population that they used as reference.

Page 5 686-87: The seropositive patients who presented cardiac alterations were classified as chronic Chagas cardiomyopathy (CCC, n= 1,707) and asymptomatic (ASY, n= 1,183). The sex distribution for the entire Latin American population studied was 61.4% female and 38.6% male.

Does the sex distribution is significantly different between the two groups on the whole cohort and in each population.

Page 7 153-154: Evaluation of functionality of the three SNPs analyzed was performed with the online software HaploReg v4.1.

The authors need to use some tools that are more appropriate as remap which is also using the Encode library. However, these data were cured before to be included in Remap.

Page 9: Regarding infection levels, three polymorphisms were associated in the Colombian population and none of them in the Argentinian population. Then authors performed a meta-analysis on these two populations. At this level authors indicated that the polymorphism IL18 rs360719 is associated.

In the same way the authors do not underlined the results obtained in their meta-analysis for rs1946518. It is not consistent.

I will really appreciate if the authors can compare the frequencies of these SNPs between the seronegative individuals extracted from the Colombian population vs seronegative individuals extracted from the Argentinian population. It may provide arguments to better understand their meta-analysis data.

Page 10: for chronic disease the authors detect one association in the Columbian population that is not confirmed in the Argentinian population or in the Bolivian population. In the previous study done on a Brazilian cohort no association was detected too. On the Brazilian population significant association was detected when the authors compared the moderate CCC vs the severe CCC.

I invite the authors to compare severe CCC vs moderate CCC in their Colombian, Argentinian and Bolivian population. 

Without this analysis the sentence “Regarding genetic control of the chronic Chagas cardiomyopathy, no association was detected in a well-powered cohort” is too strong.

Page 11-12: 

Authors wrote sentences in their discussion

These differences could be due to the complex genetic structure of Latin American individuals, reflected by the recent admixture among Native American, European, and West African source populations 

However, these results were not validated in the Bolivian and Argentinian cohorts. These discrepancies in the results could be due to the genetic heterogeneity among the study cohorts (Bryc, et 277 al. 2010). 

These two sentences describe a very important point of this manuscript. I fully agree. I therefore invite the authors to go further in this reflection. Do we have information on the origin of these population in terms of origin and migration. Moreover, the reference needs to be properly formatted.

Minor corrections:

Page 4 64-65: Polymorphism in genes encoding cytokines may influence the level of cytokines production and, consequently, cause different immunological responses [10, 11].

The selected references are not appropriate it‘s is better to indicate reference on Chagas disease 

Page 4 68-70: Previous genetic studies performed in a Colombian and Brazilian cohort found associations between variants of IL18 gene and the predisposition to T. cruzi infection and chronic Chagas cardiomyopathy [13, 14]. 

The authors need to respect the time of publication. Brazil study came out first.

Finally, the authors need to correct the manuscript that includes several typing errors

**Summary and General Comments**

Reviewer #1: The manuscript presented by the authors may definitively help to continue advancing towards a better understanding of the genetic causes leading to cardiomyopathy in Chagasic patients. Due to the power limitations of the studies performed to date, it is crucial to confirm the available genetic associations in larger and independent populations. Because of this, collaborative efforts like the one reported here are required. If the authors could address properly the concerns describe above, I have no doubt that this could be a highly valuable manuscript.

Reviewer #2: (No Response)

PLOS authors have the option to publish the peer review history of their article (what does this mean?). If published, this will include your full peer review and any attached files.

Reviewer #1: No

Reviewer #2: No

---

## [Decision Letter · Decision Letter 1]

21 Oct 2019

Dear Dr Strauss,

We are pleased to inform you that your manuscript, "Association of IL18 genetic polymorphisms with Chagas disease in Latin American populations", has been editorially accepted for publication at PLOS Neglected Tropical Diseases.

Before your manuscript can be formally accepted and sent to production you will need to complete our formatting changes, which you will receive in a follow up email. Please note: your manuscript will not be scheduled for publication until you have made the required changes.

IMPORTANT NOTES

* Copyediting and Author Proofs: To ensure prompt publication, your manuscript will NOT be subject to detailed copyediting and you will NOT receive a typeset proof for review. The corresponding author will have one final opportunity to correct any errors when sent the requests mentioned above. Please review this version of your manuscript for any errors.

* If you or your institution will be preparing press materials for this manuscript, please inform our press team in advance at plosntds@plos.org. If you need to know your paper's publication date for media purposes, you must coordinate with our press team, and your manuscript will remain under a strict press embargo until the publication date and time. PLOS NTDs may choose to issue a press release for your article. If there is anything that the journal should know, please get in touch.

*Now that your manuscript has been provisionally accepted, please log into EM and update your profile. Go to http://www.editorialmanager.com/pntd, log in, and click on the "Update My Information" link at the top of the page. Please update your user information to ensure an efficient production and billing process.

*Note to LaTeX users only - Our staff will ask you to upload a TEX file in addition to the PDF before the paper can be sent to typesetting, so please carefully review our Latex Guidelines [http://www.plosntds.org/static/latexGuidelines.action] in the meantime.

Best regards,

Igor C. Almeida

Associate Editor

Ana Rodriguez

Deputy Editor

Reviewer's Responses to Questions

Key Review Criteria Required for Acceptance?

Methods

-Are the objectives of the study clearly articulated with a clear testable hypothesis stated?

-Is the study design appropriate to address the stated objectives?

-Is the population clearly described and appropriate for the hypothesis being tested?

-Is the sample size sufficient to ensure adequate power to address the hypothesis being tested?

-Were correct statistical analysis used to support conclusions?

-Are there concerns about ethical or regulatory requirements being met?

Reviewer #1: No additional comments.

Reviewer #2: Are the objectives of the study clearly articulated with a clear testable hypothesis stated?

YES

-Is the study design appropriate to address the stated objectives?

YES

-Is the population clearly described and appropriate for the hypothesis being tested?

YES

-Is the sample size sufficient to ensure adequate power to address the hypothesis being tested?

YES

-Were correct statistical analysis used to support conclusions?

YES

-Are there concerns about ethical or regulatory requirements being met?

YES

Results

-Does the analysis presented match the analysis plan?

-Are the results clearly and completely presented?

-Are the figures (Tables, Images) of sufficient quality for clarity?

Reviewer #1: No additional comments.

Reviewer #2: -Does the analysis presented match the analysis plan?

YES

-Are the results clearly and completely presented?

YES

-Are the figures (Tables, Images) of sufficient quality for clarity?

YES

Conclusions

-Are the conclusions supported by the data presented?

-Are the limitations of analysis clearly described?

-Do the authors discuss how these data can be helpful to advance our understanding of the topic under study?

-Is public health relevance addressed?

Reviewer #1: No additional comments.

Reviewer #2: -Are the conclusions supported by the data presented?

YES

-Are the limitations of analysis clearly described?

YES

-Do the authors discuss how these data can be helpful to advance our understanding of the topic under study?

YES

-Is public health relevance addressed?

YES

Editorial and Data Presentation Modifications?

Reviewer #1: No additional comments.

Reviewer #2: I rewiewed the original submission and did several comments (major and minor ones)

the authors took in considerations all the remarks I made

they answered to all these queris and their responses were appropriates

In this way, the quality of the manuscript has been seriously improved

Summary and General Comments

Reviewer #1: The manuscript has been substantially improved. All my concerns have been solved.

Reviewer #2: NA

PLOS authors have the option to publish the peer review history of their article (what does this mean?). If published, this will include your full peer review and any attached files.

Do you want your identity to be public for this peer review?

 For information about this choice, including consent withdrawal, please see our Privacy Policy.

Reviewer #1: No

Reviewer #2: No

---

## [Editor Report · Acceptance letter]

13 Nov 2019

Dear Dr Strauss,

We are delighted to inform you that your manuscript, "Association of *IL18* genetic polymorphisms with Chagas disease in Latin American populations," has been formally accepted for publication in PLOS Neglected Tropical Diseases.

Best regards,

Serap Aksoy

Editor-in-Chief

Shaden Kamhawi

Editor-in-Chief
